# Factors Affecting the Detection of Hexavalent Chromium in Cr-Contaminated Soil

**DOI:** 10.3390/ijerph19159721

**Published:** 2022-08-07

**Authors:** Mingtao Huang, Guoyu Ding, Xianghua Yan, Pinhua Rao, Xingrun Wang, Xiaoguang Meng, Qiantao Shi

**Affiliations:** 1School of Chemistry and Chemical Engineering, Shanghai University of Engineering Science, Shanghai 201620, China; 2State Key Laboratory of Environmental Criteria and Risk Assessment, Chinese Research Academy of Environmental Sciences, Beijing 100012, China; 3Beijing Key Laboratory of Aqueous Typical Pollutants Control and Water Quality Safeguard, School of Environment, Beijing Jiaotong University, Beijing 100044, China; 4Stevens Institute of Technology, Hoboken, NJ 07030, USA

**Keywords:** hexavalent chromium, alkali digestion, Cr-contaminated soil, chromite ore processing residue, extraction

## Abstract

The alkali digestion pretreatment method in the United States Environmental Protection Agency (USEPA) Method 3060A could underestimate the content of Cr(VI) in Cr-contaminated soils, especially for soils mixed with chromite ore processing residue (COPR), which leads to a misjudgment of the Cr(VI) level in soils after remediation, causing secondary pollution to the environment. In this study, a new pretreatment method to analyze Cr(VI) concentration in contaminated soils was established. The impacts of soil quality, particle size, alkali digestion time and the rounds of alkali digestion on Cr(VI) detection in contaminated soils was explored and the alkali digestion method was optimized. Compared with USEPA Method 3060A, the alkaline digestion time was prolonged to 6 h and multiple alkali digestion was employed until the amount of Cr(VI) in the last extraction was less than 10% of the total amount of Cr(VI). Because Cr(VI) in COPR is usually embedded in the mineral phase structure, the hydration products were dissolved and Cr(VI) was released gradually during the alkaline digestion process. The amount of Cr(VI) detected showed high correlation coefficients with the percentage of F1 (mild acid-soluble fraction), F2 (reducible fraction) and F4 (residual fraction). The Cr(VI) contents detected by the new alkaline digestion method and USEPA Method 3060A showed significant differences for soil samples mixed with COPR due to their high percentage of residual fraction. This new pretreatment method could quantify more than 90% of Cr(VI) in Cr-contaminated soils, especially those mixed with COPR, which proved to be a promising method for Cr(VI) analysis in soils, before and after remediation.

## 1. Introduction

Chromium salt is an important industrial raw material, which is often used in chemical, machinery, electronics and other industries, and plays an important role in the construction and development of the national economy [1]. However, large amounts of chromite ore processing residue (COPR) from the production of chromium salt are not properly treated, which makes the nearby soil vulnerable to pollution by hexavalent chromium (Cr(VI)) in the COPR [2,3]. Hexavalent chromium has been classified as a class A carcinogen by the United States Environmental Protection Agency and possesses strong oxidation ability and strong mobility, which is potentially harmful to the environment and human health [4].

At present, chemical reduction has become the most widely used remediation method for Cr(VI)-contaminated soil due to its high efficiency, strong adaptability and low cost [5]. Remedial agents include ferrous sulfate, calcium polysulfide, sodium metabisulfite and so on [6,7,8,9,10,11,12,13]. However, in some remedial engineering projects, it is found that a relatively high concentration of Cr(VI) was detected in the soil treated with excessive reductants after being stored in the field for a period of time and the color of the treated soil turned yellow again, causing serious secondary pollution [14]. That might be because the oxidation rate of the reducing agent exposed to air is fast and the release speed of Cr(VI) embedded in COPR is slow, so there is not sufficient time for the reductant to react with Cr(VI), leading to Cr(VI) release after storage for a period of time, although large amounts of excessive reductant are applied [15,16,17].

Moreover, the conventional method has some problems in detecting Cr-contaminated soil in the actual site, which cannot extract all the Cr(VI) in COPR, since Cr(VI) is usually embedded in the mineral phase structure, such as silicate, which is difficult to extract, leading to a large deviation in the detection results [16,18,19,20]. Dermatas et al. [21] discovered that the method of X-ray absorption near the edge structure in spectroscopy (XANES) and the pre-treatment method of USEPA 3060A had a great difference in the detection of Cr(VI). Malherbe et al. [22] pointed out that, compared with the XANES analysis method, the conventional method underestimated the Cr(VI) content in the Cr-contaminated soil. These studies suggest that there are certain limits in the extraction of Cr(VI) from soil by the conventional alkaline digestion method, which leads to an inaccurate detection result of Cr(VI). Although XANES is a powerful tool that can distinguish between Cr(III) and Cr(VI), unfortunately, synchrotron radiation facilities are rare and expensive, which could not be used frequently during remedial project. Therefore, it is necessary to develop an appropriate and reliable method to extract Cr(VI) from Cr-contaminated soils, especially soils mixed with COPR.

The purpose of this study is to improve the alkaline digestion method and establish a new pretreatment method to analyze Cr(VI) concentration in contaminated soils. We explored the impacts of soil quality, particle size, alkali digestion time and the rounds of alkali digestion on Cr(VI) detection in contaminated soils and optimized the alkali digestion method; investigated the alkali digestion mechanism through Cr species characterization and XRD analysis before and after digestion; and validated the accuracy of the new pretreatment method with different contaminated soils.

## 2. Materials and Methods

### 2.1. Materials

Chemicals including potassium dichromate (K_2_Cr_2_O_7_), anhydrous magnesium chloride (MgCl_2_), sodium hydroxide (NaOH), anhydrous sodium carbonate (Na_2_CO_3_), dipotassium hydrogen phosphate (K_2_HPO_4_), potassium dihydrogen phosphate (KH_2_PO_4_), nitric acid (HNO_3_), acetone (C_3_H_6_O), 1,5-diphenylcarbazide (C_13_H_14_N_4_O), hydrofluoric acid (HF), perchloric acid (HClO_4_), sulfuric acid (H_2_SO_4_), hydrochloric acid (HCl), phosphoric acid (H_3_PO_4_), ammonium acetate (NH_4_OAc), hydroxylamine hydrochloride (NH_2_OH·HCl) and acetic acid (HAc) were of analytical grade and purchased from Sinopharm in China. Ultra-pure water was prepared by Milli-Q unit (Millipore, Burlington, MA, USA).

### 2.2. Soils

Four soil samples were collected. Soil A was collected from a COPR heap after remediation in Gansu, China, and its color turned to yellow again after a period of storage. The main composition of soil A was COPR after remediation (rCOPR). Soils B, C and D were collected from a COPR heap in Henan, China, the main composition of which was COPR, soil mixed with COPR and soil, respectively. Soil samples were taken from 20 to 50 cm depth. After the removal of large stones and debris, the soil samples were stored in polyethylene bags and transported to laboratory. They were dried under natural conditions and then placed in an oven at 90 °C for three days. The dried soils were ground and mixed thoroughly and then passed through 2 mm sieves. The texture, pH value, total Cr content, Cr(VI) content and residual fraction of Cr by BCR method in the soils were determined. The physical and chemical properties of the soil samples are shown in Table 1.

### 2.3. Experimental Methods

#### 2.3.1. Alkaline Digestion Experiment

The conventional alkaline digestion method is the most widely used extraction procedure for Cr(VI) analysis [23]. Briefly, approximately 2.5 g of soil was weighed and placed into a 250 mL conical flask. Then, 50 mL of the digestion solution consisting of 0.5 M NaOH/0.28 M Na_2_CO_3_ was added into the conical flask along with 400 mg of MgCl_2_ and 0.5 mL of a 1.0 M K_2_HPO_4_/KH_2_PO_4_. Finally, the solution was heated for 1 h at 95 °C by using the magnetic stirring heater.

As COPR was difficult to digest, soil A was chosen to investigate the impact factors of alkali digestion on Cr(VI) analysis, including digestion time, soil quality, particle size and the rounds of digestion. The alkali digestion time was set to 1, 2, 3, 4, 5, 6 and 7 h, respectively, to study the influence of time. Then, 0.5, 1.5 and 2.5 g of soils samples were digested for 1, 3 and 6 h, respectively, to examine the effect of soil quality on digestion. The soils were ground and passed through 100-, 200- and 400-mesh sieve and the meshes of soils <100, 100–200, 200–400 and <400 were collected and used to study the influence of particle size. After the digestion time, soil quality and particle size for digestion were determined, multiple alkali digestion was conducted. Briefly, after the first round of digestion, the residue of soil A was separated with a 0.45 μm glass fiber membrane (Jinteng, Tianjin, China) and dried with pure nitrogen. Then, the same operation was repeated for four rounds, respectively. The extract obtained in each round was analyzed by inductively coupled plasma-optical emission spectrometry (ICP−OES) for Cr(VI). Three parallel samples were prepared for each experiment.

Previous study showed that the combination of Na_2_CO_3_ and NaOH was the most effective extraction solution compared to other commonly used extractant such as phosphate buffer and hydroxide solution. Metal ions such as Ba(II) and Pb(II) form an insoluble chromate precipitate as carbonate or hydroxide while Cr(VI) goes into the solution [24,25]. The heating temperature 90–95 °C ensured the best extraction rate and prevented the oxidation of Cr(III) into Cr(VI) due to the high heating temperature. A phosphate buffer with Mg^2+^ was added into the alkaline extraction solution to prevent Cr(III) oxidation [23]. Therefore, in this study the extractant and temperature were maintained the same as the conventional method.

#### 2.3.2. New Pretreatment Method Establishment and Validation

According to the results of alkaline digestion experiment, soil quality, alkali digestion time, particle size and the rounds of alkali digestion were determined and defined as the “new pretreatment method”. Soil A, B, C and D were digested by the new pretreatment method and the conventional methods and then the extract was analyzed by ICP-OES for Cr(VI). In comparison, the content of Cr(VI) in the four soil samples was also analyzed by XANES.

During alkali digestion, conical bottles were sealed with polyethylene film to ensure that they were carried out without oxygen and three parallel samples were prepared for each experiment.

### 2.4. Analytical Methods

Soil texture and pH were determined based on the method of “Soil Agrochemical Analysis Method” [26]. The total content of Cr in the soil was pretreated by USEPA Method 3051 A [27]. The Cr(VI) concentrations and the dissolved total Cr concentrations after pretreatment were measured by ICP−OES [28]. BCR method was used to examine the Cr fractions in the soils [29]. Three parallel samples were conducted for each sample for the above analysis and their average value was taken.

The mineral composition in the soil samples was determined by X-ray diffraction (XRD, Bruker D8 Advance) employing Cu-Kα radiation in a range of 2θ from 5° to 90° with speed of 5°/min.

Trivalent chromium and Cr(VI) in soil samples were characterized by Cr K-edge (5989 eV) XAS. The XAS spectra were collected at beamline 6 BM at the National Synchrotron Light Source II (NSLS-II, Brookhaven National Laboratory, Upton, NY, USA). An energy range of −200 to 13 k, relative to the Cr K-edge, was used to acquire the spectra with a 4-element vortex silicon-drift detector. The XAS spectra of chromite nitrate (Cr(NO_3_)_3_) and sodium dichromate (Na_2_Cr_2_O_7_) were also collected as reference.

The X-ray absorption near-edge structure (XANES) linear combination fitting (LCF) analysis was conducted using the ATHENA program in the Demeter computer package [30,31]. The LCF fitting ranged from −30 to 80 eV relative to the Cr K-edge. During the LCF fitting, no energy shift was allowed and the total weight was constrained to 1. The error of overall fitness was determined using R ($R=\Sigma {\left(\chi_{\mathrm{data}}-\chi_{\mathrm{fit}}\right)}^{2}/\Sigma {\left(\chi_{\mathrm{data}}\right)}^{2}{}_{\;}$, R < 0.05).

## 3. Results and Discussion

### 3.1. The Effect of Alkali Digestion on Cr(VI) Analysis

#### 3.1.1. The Effect of Digestion Time

Figure 1 shows the effect of alkali digestion time on the detection of Cr(VI) in soil A. The alkali digestion time has a significant effect on the detection of Cr(VI) in soil A. The amount of Cr(VI) in soil A increased with an increase in digestion time, from 185.01 mg/kg in 1 h to 420.52 mg/kg in 7 h. There are some problems in the determination of Cr(VI) in soil A using the conventional method [22]. That might be because COPR contains many hydration products, which were thermodynamically unstable in aqueous environments and at ambient temperature, such as hydrogarnet and hydrocalumite [21,32], the structure of the hydration products was damaged continuously so that more Cr(VI) was released as the digestion time increased. The amount of Cr(VI) detected in soil A increased rapidly in the first 5 h, which changed slowly and tended to be gentle when the digestion time was longer than 5 h. It can be seen that the detection of Cr(VI) reached a stable level when the digestion time reached 6 h and the average amount of Cr(VI) increased by 4.12% and 2.42% from 5 to 6 h and 6 to 7 h, respectively. Because more than 97% of Cr(VI) was extracted in 6 h digestion, 6 h digestion time was chosen in the new method.

#### 3.1.2. The Effect of Soil Quality

Figure 2 shows the amount of Cr(VI) detected with different quantities of soil A digested for 1, 3 and 6 h, respectively. It can be seen that the quantity of soil sample and digestion time has a great impact on the amount of Cr(VI) detected in soil A. When the quantity of soil A was 0.5 g, the amount of Cr(VI) detected at different digestion times was comparable, although the average of that for 6 h digestion was the highest. When the quantity of soil A for digestion was 1.5 g and 2.5 g, the amount of Cr(VI) detected increased with the digestion time. This is consistent with the results of the digestion time effect experiment. When the soil amount was 0.5 g, the average amounts of Cr(VI) detected at 3 h and 6 h were lower than those when the soil amount was 1.5 g and 2.5 g. That might be because that real soil is a complex ecosystem, comprising minerals, organic matter, microorganisms and other solid components [33]. That is to say, soil is heterogeneous and for lesser quantities of soil, it was difficult to represent the properties of soil, which increased the contingency of the experiment. Therefore, the 2.5 g soil sample was selected for digestion to determine the amount of Cr(VI) in soil, which is the same as the conventional alkaline digestion method.

#### 3.1.3. The Effect of Particle Size

The effect of particle size on Cr(VI) detection in soil A is shown in Figure 3. The amounts of Cr(VI) in soil A with different particle sizes of <100, 100–200, 200–400 and <400 mesh were 410.33 mg/kg, 570.33 mg/kg, 322.14 mg/kg and 316.10 mg/kg, respectively. The amount of Cr(VI) detected in soil A with the mesh of 100–200 was the highest. The reason might be that soil A mainly comprises COPR, which is difficult to break [34,35]. Therefore, the mesh of <100 is the best choice for alkali digestion.

#### 3.1.4. The Effect of Multiple Alkali Digestion

According to the above experiments, the 2.5 g sample with a particle size less than 100 mesh and 6 h digestion time was chosen for alkaline digestion and then multiple alkali digestion was conducted. As shown in Figure 4, as the rounds of alkali digestion increased, the amount of Cr(VI) in soil A decreased gradually, and Cr(VI) can still be detected after conducting digestion for six times. Hexavalent chromium in COPR is mainly embedded in silicate minerals, which are not easily dissolved during alkali digestion [36]. In the last two alkali digestion processes, the amount of Cr(VI) in soil A decreased to 10.35 mg/kg and 3.65 mg/kg, accounting for 1.25% and 0.44% of the total amount of Cr(VI) from the six digestion rounds, which means that the amount of Cr(VI) detected by the first four rounds of digestion accounted for more than 98% of the total amount. Considering detection accuracy and time consumption, multiple alkali digestion was employed to detect the amount of Cr(VI) in the soil until the amount of Cr(VI) in the last extraction was less than 10% of the total amount of Cr(VI).

### 3.2. New Method Establishment and Validation

#### 3.2.1. New Method Validation with Different Cr-Contaminated Soils

The amounts of Cr(VI) detected by the conventional method and the new method are shown in Figure 5. Using the conventional method, the amount of Cr(VI) detected in soils A, B, C and D was 157.41 mg/kg, 5295.68 mg/kg, 2786.01 mg/kg and 400.13 mg/kg, respectively. Using the new method, the amount of Cr(VI) detected in soils A, B, C and D increased from 410.33 mg/kg, 6364.51 mg/kg, 2934.53 mg/kg and 415.12 mg/kg for digestion once to 761.51 mg/kg, 7452.05 mg/kg, 3135.52 mg/kg and 461.01 mg/kg for four repetitions of digestion, respectively. Because soil A was mainly composed of rCOPR, ettringite is the main residual fraction in rCOPR. During alkali digestion, Na_2_CO_3_ gradually eroded ettringite and more Cr(VI) can be released by using the new method, so that the amount of Cr(VI) detected was about 3.8-times of that by the conventional method [37]. Soils B and C were mainly composed of COPR, in which approximately 75–85% Cr in the COPR is present as Cr(III) and 20–25% is present as Cr(VI). The Cr(VI) in the COPR is mainly in hydrogarnet and hydrocalumite and about 1% presents in ettringite [38]. Hydrogarnet and hydrocalumite are unstable hydration products, which can be dissolved and form relatively stable gibbsite in the process of alkali digestion [21,32].

Compared with the conventional method, the amount of Cr(VI) in soils A, B, C and D, measured by the new method, was higher, which increased by 383.75%, 40.72%, 12.54% and 15.21%, respectively. Soils A, B, C and D were digested for 4, 5, 5 and 3 rounds by the new method and the amount of Cr(VI) increased by 85.58%, 17.09%, 6.85% and 11.05% compared to one-time digestion, respectively. Therefore, it is necessary to adopt the new method and carry out multiple alkali digestion processes for Cr-contaminated soil analysis, especially for those containing COPR and rCOPR, such as soils A, B and C. Because there is still a certain amount of insoluble Cr(VI) in soil D, the amount of Cr(VI) detected by the new method was 15.21% higher than that by the conventional method.

In summary, the amount of Cr(VI) in different soil samples detected by the new method can provide more accurate results, which is essential for the evaluation of Cr(VI) level in contaminated soil and remediation effect after treatment.

#### 3.2.2. Fraction Analysis of Cr in the Cr-Contaminated Soil

The fraction of Cr in the soils by BCR analysis is shown in Figure 6. According to the method, Cr in the contaminated soil can be categorized into four fractions: mild acid-soluble fraction (F1), reducible fraction (F2), oxidizable fraction (F3) and residual fraction (F4). In the four soils, F4 in soils A, B, C and D accounted for 66.34%, 47.08%, 45.33% and 14.59%, respectively. Since soils A, B and C contained rCOPR or COPR, their residual fractions were much higher than that of soil D. The amount of Cr(VI) detected by the new method and the conventional method showed high correlation coefficients with F1, F3 and F4 (Table 2), and the correlation coefficients with F3 were 0.9726 and 0.9281, respectively, higher than those with F1 and F4. Thus, Cr(VI) mainly presents in mild acid-soluble, oxidizable fraction and residual fraction.

F1, F2 and F3 are the main bioavailable forms, in which F1 and F2 are sensitive to environmental changes, easy to migrate and transform and easy to release under acidic conditions, and F3 has strong binding ability to soil organic matter. Therefore, it is easy for plants to extract and utilize them, while, in comparison, F4 is more stable in nature [39]. However, previous study has shown that the residual fraction of Cr could be transformed into mild acid-soluble fraction, reducible fraction and oxidizable fraction of Cr during alkali digestion [32]. For soil A, that the amount of Cr(VI) detected by the new method and the conventional method exhibited significant differences, which might be due to its high content of residual fraction. The release of Cr(VI) in residual fraction is slow, leading to inaccuracy in the conventional method. While for soil D, the residual fraction only accounts for 14.59%, the amount of Cr(VI) detected by the new method increased by 15.21% compared to the conventional method.

#### 3.2.3. Spectroscopic Analysis of Soils

(1) XRD Analysis

The XRD patterns of soil A with different multiple alkali digestion processes are shown in Figure 7. The main crystallization peak in soil A was ettringite (3CaO·Al_2_O_3_·3CaSO_4_·32H_2_O, 23.07°, 29.35°, 39.34° and 48.50°), followed by hydrogarnet (Ca_3_(Al,Fe)_2_(OH)_12_, 39.34°), hydrocalumite (Ca_2_(Al,Fe)(OH)_7_·3H_2_O, 20.89°), gibbsite (Al(OH)_3_·3H_2_O, 32.02°), chromite (FeCr_2_O_4_, 35.97°, 43.12° and 57.42°), maghemite/magnetite (Fe_2_O_3_/Fe_3_O_4_, 35.97°) and quartz (SiO_2_, 26.61°). The crystallization peak corresponding to hydrogarnet, hydrocalumite and ettringite gradually decreased with the increase in alkali digestion round. Some studies showed that hydrogarnet and hydrocalumite were hydration products, which were thermodynamically unstable in aqueous environments and at ambient temperature [16,40] and ettringite can be gradually eroded by Na_2_CO_3_ [38]. During the multiple digestion process, the Cr(VI) embedded in ettringite and other mineral phase structures gradually released with the dissolution of minerals [41].

(2) XANES Analysis

Cr K-edge XANES spectra were performed on the samples to analyze the change in Cr(VI) content in soils. Table 3 shows the ratio of Cr(VI) and Cr(III) to the total Cr mass fraction in XANES analysis. All fitting results of R were <0.05 and the LCF fitting results are in line with the fitting standard. In XANES spectra, a definite pre-edge of Cr(VI) appears around 5993.5 eV and the height as well as area of the pre edge are related to the total Cr content in Cr(VI) [14].

Quantitative LCF results for Cr species in soil samples are summarized in Table 3. Except that the original soil D sample contained 73.5% Cr(III) and 26.5% Cr(VI), the Cr(VI) contents were less than 0.1% in other soil samples. In raw COPR, approximately 75–85% of the chromium is present as Cr(III) and 15–25% is present as Cr(VI). The Cr(VI) is present in hydrogarnet, hydrocalumite and ettringite, among which hydrogarnet has the highest content and ettringite accounts for less than 1%. COPR is usually stored in the field for a long time. As hydrogarnet and hydrocalumite are unstable, the minerals gradually dissolved and transformed into stable gibbsite in the long-term weathering process, making the insoluble Cr(VI) gradually release and transform into soluble Cr(VI) [38,42]. Therefore, the chromium in most samples is mainly Cr(III), which makes the proportion of Cr(VI) in the soils relatively low and falls below the detection limit of XANES.

The amount of Cr(VI) detected by the conventional method, new method and XANES in soil D was 400.13 mg/kg, 461.01 mg/kg and 501.47 mg/kg, respectively. This result is consistent with the finding by Malherbe J [22], indicating that the USEPA method 3060A underestimated the content of Cr(VI) in soil, especially those mixed with COPR. Compared to the conventional method, the amount of Cr(VI) measured by the new method established by this study is closer to that by XANES, which accounts for 91.9% of the total content. This new method is proved to be a more promising method for Cr(VI) analysis in soils, especially for those mixed with COPR (Figure 8).

## 4. Conclusions

As the conventional alkaline digestion method underestimated the content of Cr(VI) in Cr-contaminated soils, a new pretreatment method to analyze Cr(VI) concentration in contaminated soils was established in this study. The new method extended the alkali digestion time from 1 to 6 h on the basis of the conventional method and multiple alkali digestion processes were carried out until the amount of Cr(VI) in the last extraction was less than 10% of the total amount of Cr(VI). This new pretreatment method was validated by different soils, which mainly comprised of rCOPR, COPR and COPR mixed soil, and soil. Chromium(VI) mainly presents in F1, F2 and F4, and its amount detected by the conventional and new pretreatment methods showed high correlation coefficients with the percentage of F1, F2 and F4. For the soils mixed with COPR, the amount of Cr(VI) detected by the new and conventional methods exhibited significant differences due to its high content of residual fraction. In summary, more than 90% of Cr(VI) can be quantified by this new method, showing that it is a promising method for Cr(VI) analysis in contaminated soil before and after remediation, especially for those mixed with COPR, to avoid the misjudgment of remediation effectiveness.

## Figures and Tables

**Figure 1 ijerph-19-09721-f001:**
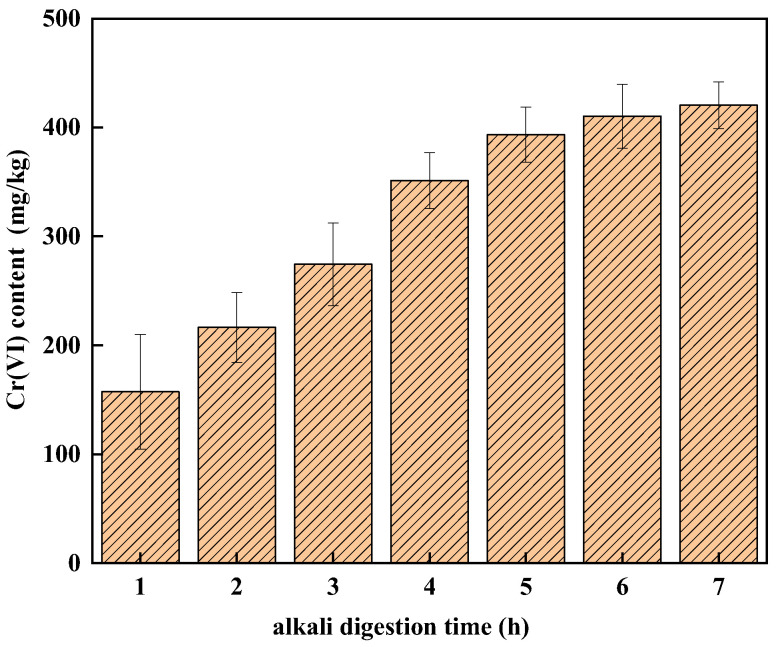
The effect of digestion time on the detection of Cr(VI) in soil A.

**Figure 2 ijerph-19-09721-f002:**
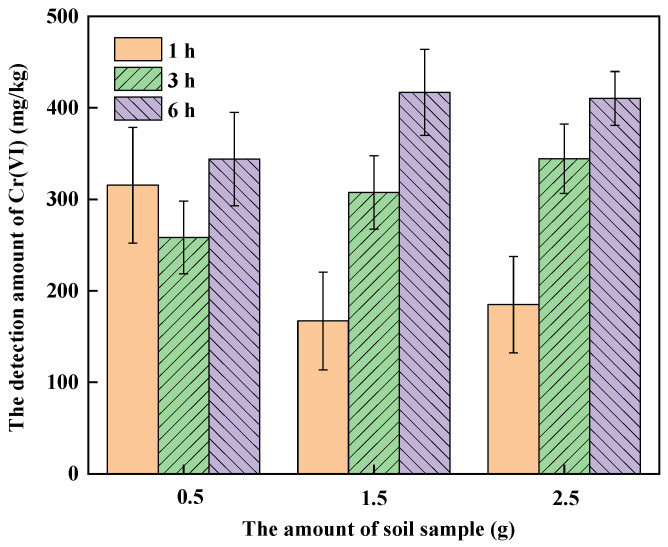
The effect of digestion time and soil quality on the detection of Cr(VI) in soil A.

**Figure 3 ijerph-19-09721-f003:**
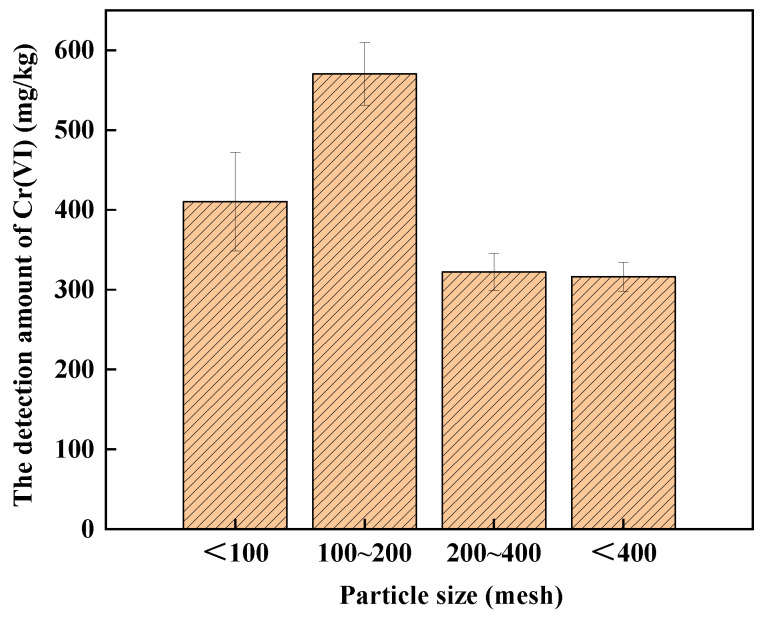
The effect of particle size on the detection of Cr(VI) in soil A.

**Figure 4 ijerph-19-09721-f004:**
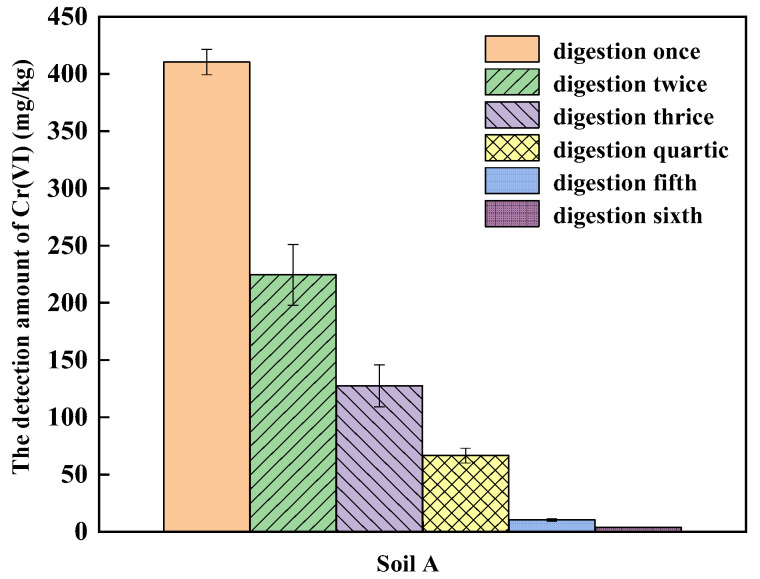
The effect of multiple alkali digestion on the detection of Cr(VI) in soil A.

**Figure 5 ijerph-19-09721-f005:**
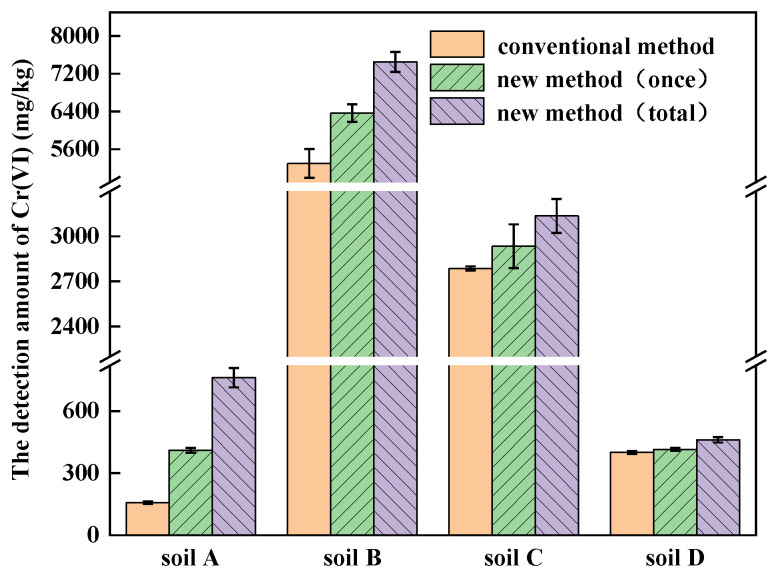
Validation of different Cr-contaminated soils with new method.

**Figure 6 ijerph-19-09721-f006:**
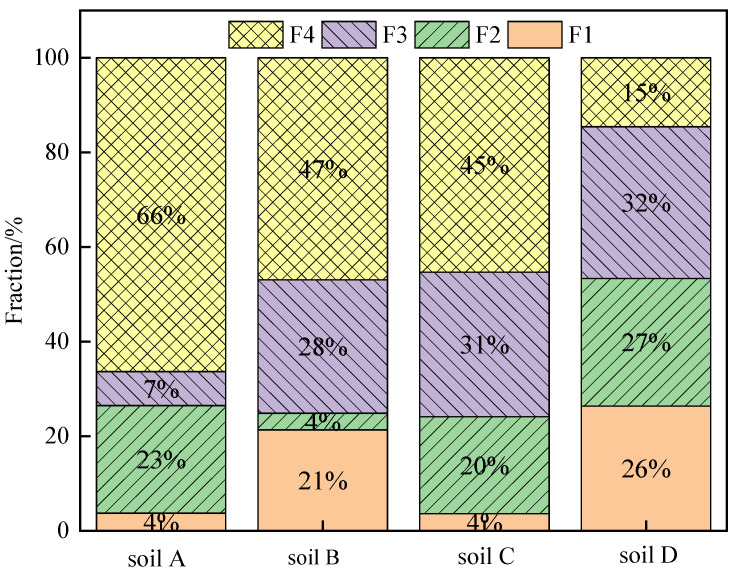
Fractionation of Cr in the soils using BCR method.

**Figure 7 ijerph-19-09721-f007:**
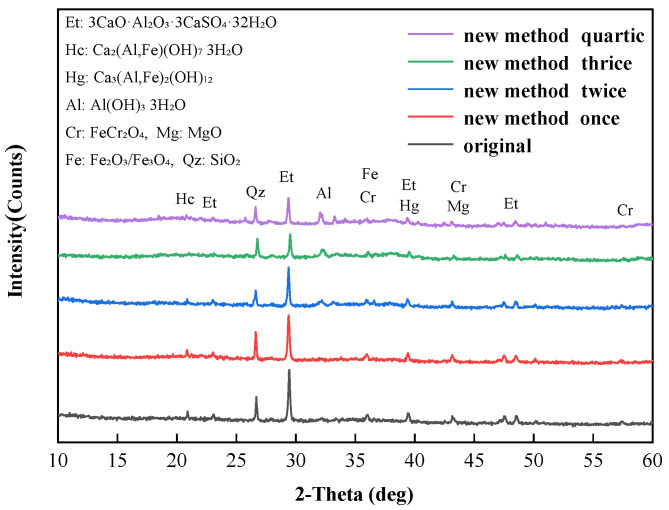
The XRD patterns of soil A at different alkali digestion rounds.

**Figure 8 ijerph-19-09721-f008:**
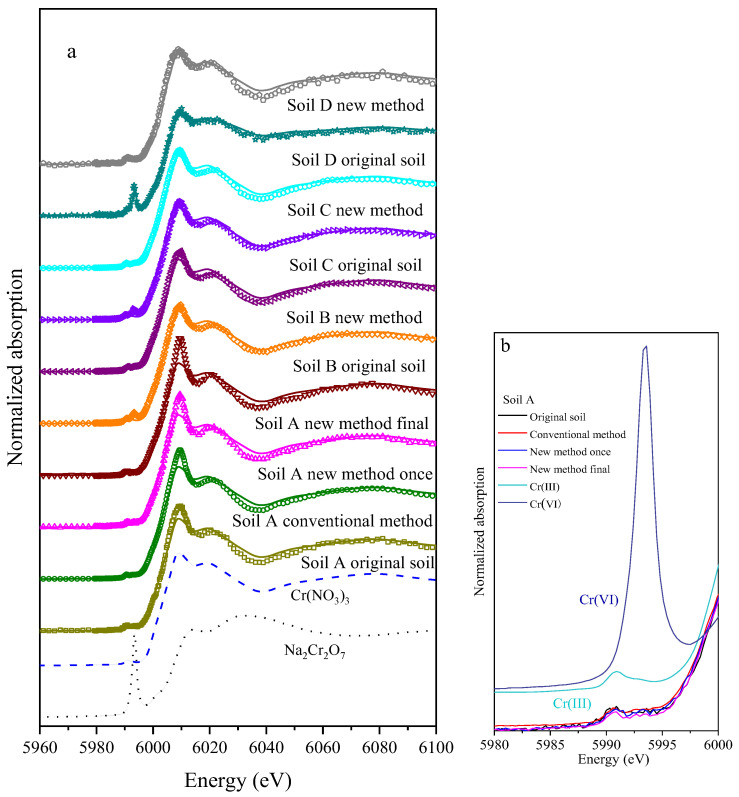
XANES LCF fitting for Cr species in the soil samples and standards (**a**), pre-edge of samples in the soil A (**b**).

**Table 1 ijerph-19-09721-t001:** Physiochemical properties of soils samples.

Sample	Soil Texture	pH	Cr(VI) Content/(mg/kg)	Total Cr Content/(mg/kg)	The Residual Fraction of Cr by BCR Method
Soil A	Sandy loam	9.19 ± 0.19	185.01 ± 15.33	2.12 × 10^4^ ± 1110.38	66.33%
Soil B	Sandy loam	11.58 ± 0.23	5295.68 ± 253.21	5.43 × 10^4^ ± 3865.25	75.50%
Soil C	Sandy loam	11.41 ± 0.35	2786.01 ± 112.35	1.08 × 10^4^ ± 566.32	45.33%
Soil D	silt	7.3 ± 0.33	373.13 ± 56.32	1.89 × 10^3^ ± 78.35	14.59%

Note: Cr(VI) content was measured by the conventional method.

**Table 2 ijerph-19-09721-t002:** Correlation matrix for the fraction of Cr and Cr(VI) detected by the conventional and new pretreatment method.

Element	F1	F2	F3	F4	Cr(VI)
F1	1				
F2	−0.1779	1			
F3	0.9855	−0.1670	1		
F4	0.8712	0.3279	0.8604	1	
Cr(VI) (New method)	0.9200	−0.1974	0.9726	0.7794	1
Cr(VI) (Conventional method)	0.8574	−0.2847	0.9281	0.6743	1

**Table 3 ijerph-19-09721-t003:** Quantitative LCF results for Cr species in soil samples.

Soil	Sample	XANES Fitting		
		Species Percentage (%)		R-Factor
		Cr(Ⅲ)	Cr(VI)	
Soil A	original soil	100	<0.1	0.0126309
	conventional method	100	<0.1	0.0138056
	new method once	100	<0.1	0.0178716
	new method final	100	<0.1	0.0226955
Soil B	original soil	100	<0.1	0.0062285
	new method	100	<0.1	0.0084095
Soil C	original soil	100	<0.1	0.0079396
	new method	100	<0.1	0.0073264
Soil D	original soil	73.5	26.5	0.0058493
	new method	100	<0.1	0.0093941

## Data Availability

The data presented in this study are available from the corresponding author on reasonable request.

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
