# Peer review of "Factors Affecting the Detection of Hexavalent Chromium in Cr-Contaminated Soil"

_ijerph, 2022, doi:10.3390/ijerph19159721_

Round 1
Reviewer 1 Report
The author must report the contribution of other factors, where they kept constant or maintained during this study?
The authors must use full name of abbreviation as USEPA etc.
Th authors must also add information about the Cr III residues while studying the Cr VI as this study is based om Cr detection in soil where many Cr reducing microbial strains present.
If this is a lab study then what could happened in field under natural conditions? will this method work efficiently?
Author Response
Comments from Reviewer
Open Review
( ) I would not like to sign my review report
(x) I would like to sign my review report
English language and style
( ) Extensive editing of English language and style required
(x) Moderate English changes required
( ) English language and style are fine/minor spell check required
( ) I don't feel qualified to judge about the English language and style
|
Yes |
Can be improved |
Must be improved |
Not applicable |
|
|
Does the introduction provide sufficient background and include all relevant references? |
( ) |
(x) |
( ) |
( ) |
|
Are all the cited references relevant to the research? |
( ) |
( ) |
(x) |
( ) |
|
Is the research design appropriate? |
(x) |
( ) |
( ) |
( ) |
|
Are the methods adequately described? |
( ) |
(x) |
( ) |
( ) |
|
Are the results clearly presented? |
(x) |
( ) |
( ) |
( ) |
|
Are the conclusions supported by the results? |
( ) |
(x) |
( ) |
( ) |
Response:
We would like to make a grateful acknowledgement to the reviewer for the valuable suggestions and comments on improving the manuscript. We have revised the manuscript point by point accordingly, and checked the English language carefully and made improvement.
Comments and Suggestions for Authors
Comment 1: The author must report the contribution of other factors, where they kept constant or maintained during this study?
Response: Thank you for your comment. We have added the contribution of other factors. The manuscript was revised accordingly as follows in Line 125 to 133.
Previous study showed that the combination of Na2CO3 and NaOH was the most effective extraction solution compared to other commonly used extractant such as phosphate buffer and hydroxide solution. Metal ions such as Ba(II) and Pb(II) that form an insoluble chromate precipitate as carbonate or hydroxide while Cr(VI) goes into the solution [24,25]. The heating temperature 90-95 oC ensured the best extraction rate and prevented the oxidation of Cr(III) into Cr(VI) due to the high heating temperature. A phosphate buffer with Mg2+ was added into the alkaline extraction solution to prevent Cr(III) oxidation[23]. Therefore, in this study the extractant and temperature were remained the same as the conventional method.
- USEPA, Method 3060A. In Alkaline digestion for hexavalent chromium, United States Environmental Protection Agency: Washington, DC, 1996.
- Vitale, R. J.; Mussoline, G. R.; Petura, J. C.; James, B. R., Hexavalent Chromium Extraction from Soils: Evaluation of an Alkaline Digestion Method. J Environ Qual 1994, 23, (6), 1249-1256.
- James, B. R.; Petura, J. C.; Vitale, R. J.; Mussoline, G. R., Hexavalent chromium extraction from soils: a comparison of five methods. Environ. Sci. Technol. 1995, 29, (9), 2377-2381.
Comment 2: The authors must use full name of abbreviation as USEPA etc.
Response: Thank you for your suggestion. We have provided the full name of USEPA as follows in Line 15 to 16.
“The alkali digestion pretreatment method in the United States Environmental Protection Agency (USEPA) Method 3060A could underestimate the content of Cr(VI) in Cr-contaminated soils,…”
Comment 3: The authors must also add information about the Cr III residues while studying the Cr VI as this study is based om Cr detection in soil where many Cr reducing microbial strains present.
Response: Thank you for your comment. Actually, only soluble Cr(III) in soil can be extracted by EDTA and measured directly at present, it is difficult to detect the total content of Cr(III) directly. Most total Cr(III) measurements are calculated by measuring the total Cr and Cr(VI) content, and their difference is total Cr(III) content. Because Cr(VI) was toxic to microbials, and we measured the contents of Cr(VI) by the new pretreatment method during our experiment, the results showed no difference.
Comment 4: If this is a lab study then what could happened in field under natural conditions? will this method work efficiently?
Response: Thank you for your comment. The initial purpose of developing this method is to detect Cr-contaminated soil in actual sites before and after remedial engineering project. It is found that a relatively high concentration of Cr(VI) had been detected by this new pretreatment method in the soil treated with excessive reductant compared with the conventional method in real remedial project. The treated soil turned yellow again after storage for a period of time indicating the release of Cr(VI). The new pretreatment method worked efficiently to evaluate the performance of remediation and to provide the real content of Cr(VI) in contaminated soils, especially those mixed with COPR.

Reviewer 2 Report
In the manuscript, a new pretreatment method to analyze Cr(VI) concentration in contaminated soils was established. The impacts of soil quality, particle size, alkali digestion time and the rounds of alkali digestion on Cr(VI) detection in contaminated soils was explored and the alkali digestion method was optimized. In addition, this new method could quantify more than 90% of Cr(VI) in Cr-contaminated soils. This work is interesting and meaningful for providing scientific basis for soil cadmium pollution remediation. Thus, I recommend publication in Int. J. Environ. Res. Public Health after revision. The manuscript would be excellent if the authors make some corrections as following:
1. In the abstract, give the full name of USEPA, and significance of this research was absent.
2. In introduction part, please add more comments on previous studies on analysis method of Cr.
3. Table 1, it is suitable to use scientific notation to express the total Cr content.
4. In figure 1, whether 7 h is the optimal value for alkali digestion time, and how about 8h, 9h or 10h.
5. In figure 6, is it average for this results ? how many samples ? or How many times were repeated for each sample?
6. New method establishment and validation, the authors compared the results with conventional method. Why did not analyze the reference material with certified value to validate the developed method?
7. Conclusion part can be more concise.
8. Add more recent literatures in reference list.
Author Response
Comments from Reviewer
Open Review
(x) I would not like to sign my review report
( ) I would like to sign my review report
English language and style
( ) Extensive editing of English language and style required
( ) Moderate English changes required
(x) English language and style are fine/minor spell check required
( ) I don't feel qualified to judge about the English language and style
|
Yes |
Can be improved |
Must be improved |
Not applicable |
|
|
Does the introduction provide sufficient background and include all relevant references? |
(x) |
( ) |
( ) |
( ) |
|
Are all the cited references relevant to the research? |
(x) |
( ) |
( ) |
( ) |
|
Is the research design appropriate? |
(x) |
( ) |
( ) |
( ) |
|
Are the methods adequately described? |
(x) |
( ) |
( ) |
( ) |
|
Are the results clearly presented? |
(x) |
( ) |
( ) |
( ) |
|
Are the conclusions supported by the results? |
( ) |
(x) |
( ) |
( ) |
Comments and Suggestions for Authors
In the manuscript, a new pretreatment method to analyze Cr(VI) concentration in contaminated soils was established. The impacts of soil quality, particle size, alkali digestion time and the rounds of alkali digestion on Cr(VI) detection in contaminated soils was explored and the alkali digestion method was optimized. In addition, this new method could quantify more than 90% of Cr(VI) in Cr-contaminated soils. This work is interesting and meaningful for providing scientific basis for soil cadmium pollution remediation. Thus, I recommend publication in Int. J. Environ. Res. Public Health after revision. The manuscript would be excellent if the authors make some corrections as following:
Response:
We would like to make a grateful acknowledgement to the reviewer for the valuable suggestions and comments on improving the manuscript. We have revised the manuscript point by point accordingly, and checked and corrected the spell and expression of the English language carefully.
Comment 1: In the abstract, give the full name of USEPA, and significance of this research was absent.
Response: Thank you for your suggestion. We have provided the full name of USEPA, and significance of this research as follows in line 15-16 and 30-32.
In line 15-16, “The alkali digestion pretreatment method in the United States Environmental Protection Agency (USEPA) Method 3060A could underestimate the content of Cr(VI) in Cr-contaminated soils,…”
In line 30-32, “This new pretreatment method could quantify more than 90% of Cr(VI) in Cr-contaminated soils, especially those mixed with COPR, which proved to be a promising method for Cr(VI) analysis in soils before and after remediation.”
Comment 2: In introduction part, please add more comments on previous studies on analysis method of Cr
Response: Thank you for your suggestion. We have got some comments on the conventional alkaline digestion method in line 65-67, “These studies suggest that there are certain limits in the extraction of Cr(VI) from soil by the conventional alkaline digestion method, which leads to an inaccurate detection result of Cr(VI)”. We added more comments on XANES in line 67-69 as follows.
“Although XANES is a powerful tool that can distinguish between Cr(III) and Cr(VI), unfortunately synchrotron radiation facilities are rare and expensive which could not be used frequently during remedial project.”
Comment 3: Table 1, it is suitable to use scientific notation to express the total Cr content.
Response: Thank you for your suggestion. We have changed the expression of the total Cr content using scientific notation in line 101 as follows.
Table 1 Table 1. Physiochemical properties of soils samples.
|
Sample |
Soil texture |
pH |
Cr(VI) content/(mg/kg) |
Total Cr content/(mg/kg) |
The residual fraction of Cr by BCR method |
|
Soil A |
Sandy loam |
9.19±0.19 |
185.01±15.33 |
2.12×104±1110.38 |
66.33% |
|
Soil B |
Sandy loam |
11.58±0.23 |
5295.68±253.21 |
5.43×104±3865.25 |
75.50% |
|
Soil C |
Sandy loam |
11.41±0.35 |
2786.01±112.35 |
1.08×104±566.32 |
45.33% |
|
Soil D |
silt |
7.3±0.33 |
373.13±56.32 |
1.89×103±78.35 |
14.59% |
Comment 4: In figure 1, whether 7 h is the optimal value for alkali digestion time, and how about 8h, 9h or 10h.
Response: Thank you for your comment. During experiment, we discovered that the amount of Cr(VI) detected in soil A increased rapidly in the first 5 h, and then increased slowly and tended to be gentle when the digestion time was longer than 5 h. It can be seen that the detection of Cr(VI) reached a stable level when the digestion time reached 6 h that the average detection amount of Cr(VI) increased 4.12% and 2.42% from 5 to 6 h and 6 to 7 h, respectively. Because more than 97% of Cr(VI) has been extracted in 6 h digestion, 6 h digestion time was chosen in the new method to save time for analysis, and longer digestion time than 7 h was not conducted.
Comment 5: In figure 6, is it average for this results ? how many samples ? or How many times were repeated for each sample?
Response: Thank you for your comment. In figure 6, the results were the average of three parallel samples. We have added the information in line 149-150 as follows.
“Three parallel samples were conducted for each sample for the above analysis, and their average value was taken.”
Comment 6: New method establishment and validation, the authors compared the results with conventional method. Why did not analyze the reference material with certified value to validate the developed method?
Response: Thank you for your good suggestion. The conventional method was applicable for soil contaminated with Cr(VI). However, for the contaminated soil collected from chromite ore processing residue (COPR) site is usually comprised of certain amount of COPR, containing a variety of hydration products which are difficult to be dissolved during alkaline digestion. The characteristics of soil and COPR are quite different. However, there is no COPR reference material with certified value to validate the developed method. Thus, we compared the results with conventional method and improved it for analysis of soil mixed with COPR.
Comment 7: Conclusion part can be more concise.
Response: Thank you for your comment. The conclusion part had been revised to be more concise in line 332-346 as follows.
As the conventional alkaline digestion method underestimated the content of Cr(VI) in Cr-contaminated soils, a new pretreatment method to analyze Cr(VI) concentration in contaminated soils was established in this study. The new method extended the alkali digestion time from 1 to 6 h on the basis of the conventional method, and multiple alkali digestion was carried out until the amount of Cr(VI) in the last extraction was less than 10% of the total amount of Cr(VI). This new pretreatment method was validated by different soils, which were mainly comprised of rCOPR, COPR, COPR mixed soil, and soil. Chromium(VI) mainly presents in F1, F2, and F4, and its amount detected by the conventional and new pretreatment methods showed high correlation coefficients with the percentage of F1, F2, and F4. For the soils mixed with COPR, the amount of Cr(VI) detected by the new and conventional method exhibited significant difference due to its high content of residual fraction. In summary, more than 90% of Cr(VI) can be quantified by this new method showing that it is a promising method for Cr(VI) analysis in contaminated soil before and after remediation, especially for those mixed with COPR, to avoid the misjudgment of remediation effectiveness.
Comment 8: Add more recent literatures in reference list.
Response: Thank you for your suggestion. We have added several more recent literatures in line 386-387and 401-402.
- Jiang, Y.; Yang, F.; Dai, M.; Ali, I.; Shen, X.; Hou, X.; Alhewairini, S. S.; Peng, C.; Naz, I., Application of microbial immobilization technology for remediation of Cr (VI) contamination: A review. Chemosphere 2022, 286.
- Hu, S.; Li, D.; Qin, S.; Man, Y.; Huang, C., Interference of sulfide with iron ions to the analysis of Cr(VI) by Method 3060a & Method 7196a. J. Hazard. Mater. 2020, 398-407.

Round 2
Reviewer 2 Report
The manuscript has been improved a lot, thus I recommend accept in present form.